# Community participation and technological innovation: Baseline qualitative insights to inform a five-year cohort on drone-based dengue surveillance in Malaysia

Rahmat Dapari[1,2‡], Safiyeh Tayebi[3‡], Ana Lorena Ruano[4], Timothy C. Guetterman[5], Seok Mui Wang[6], Siti Hafizah AB Hamid[7], Sohel Rahman[8], Jürgen Pilz[9], Nazri Che Dom[10], Ubydul Haque[11]*

1 Department of Community Health, Faculty of Medicine and Health Sciences, Universiti Putra Malaysia, Serdang, Malaysia, 2 Integrated Dengue Research and Development, Faculty of Medicine and Health Sciences, Universiti Putra Malaysia, Serdang, Malaysia, 3 Department of Geography, Rutgers, Rutgers University, New Brunswick, New Jersey, United States of America, 4 Center for International Health, Department of Global Health and Primary Care, University of Bergen, Bergen, Norway, 5 Department of Family Medicine, University of Michigan, Ann Arbor, Michigan, United States of America, 6 Department of Medical Microbiology & Parasitology, Faculty of Medicine, Universiti Teknologi MARA, Sungai Buloh Campus, Selangor, Malaysia, 7 Department of Software Engineering, Faculty of Computer Science & Information Technologies, University of Malaya, Kuala Lumpur, Malaysia, 8 Department of CSE, BUET, ECE Building, West Palashi, Dhaka, Bangladesh, 9 Alpen-Adria University of Klagenfurt, Klagenfurt, Austria, 10 Centre for Environmental Health and Safety Studies, Faculty of Health Sciences, Universiti Teknologi MARA, Kampus Puncak Alam, Bandar Puncak Alam, Selangor, Malaysia, 11 Rutgers Global Health Institute, Rutgers University, New Brunswick, New Jersey, United States of America

‡ These authors are joint first author on this work.
* ubydul.kth@gmail.com

## Abstract

### Background

To inform a prospective cohort study five-year automated surveillance study, this study explores households and stakeholder perceptions of using drones for mosquito breeding site surveillance as part of dengue control strategies in Selangor, Malaysia. A qualitative design identified diverse perspectives across eight high-risk localities. Data were collected through 480 in-depth interviews with household heads, from a newly established cohort of households, and six key informant interviews with public health professionals. Participants were selected using typical case and expert sampling methods to ensure representation across socioeconomic and urban heterogeneity.

### Methods

This study developed a conceptual framework integrating community-based vector control, public health technology adoption, and drone-assisted surveillance, structured into five stages: Inputs, Processes, Outputs, Outcomes, and Impacts. It was applied in Selangor, Malaysia, a dengue-endemic state, to assess the feasibility and

**Data availability statement:** All data supporting the findings of this study are fully available within this manuscript.

**Funding:** The author(s) received no specific funding for this work.

**Competing interests:** The authors have declared that no competing interests exist.

perception of drone-based interventions. Data were collected through 480 in-depth household interviews and six key informant interviews with public health experts, using semi-structured formats. Thematic analysis was conducted using Braun and Clarke's approach to identify recurring patterns across technical, organizational, and social dimensions of implementation.

## Results

Participants supported drone use when positioned as mosquito breeding site monitoring tools rather than personal surveillance. Transparent communication about purpose, data use, and operational boundaries was key to maintaining trust. Targeted use in known mosquito hotspots was preferred over random surveillance. Privacy concerns were minimal, and most households strongly supported using drones for surveillance of mosquito habitats to aid in dengue control. Many simply requested advance notice of flight schedules to stay informed and engaged.

## Conclusions

Community motivation stemmed from a sense of collective responsibility, with most participants valuing their involvement as a contribution to neighborhood well-being. Trust, transparency, and consistent communication were identified as essential for long-term engagement and the success of the project. These findings underscore the importance of aligning innovation with local social dynamics and demonstrate the value of participatory approaches in public health surveillance technologies interventions.

### Author summary

This study shows that aligning drone surveillance technologies with cultural dynamics and fostering trust through transparency and communication promotes community acceptance and sustained engagement. The research identifies key factors, including privacy, efficiency, and geographic challenges, that influence drone adoption in diverse settings, providing insights to overcome barriers and optimize implementation. Reframing drones as mosquito control tools builds community trust and highlights their global scalability for disease prevention. The research highlights the value of integrating community perspectives with technical innovation to design effective and implement context-sensitive public health surveillance technologies.

## Introduction

Technological innovation increasingly shapes the landscape of public health practice and research, offering new tools for disease surveillance, diagnosis, and control [1,2].

Among the most promising of these are unmanned aerial vehicles (UAVs) or drones, which are rapidly gaining attention because they enhance precision, efficiency, and coverage in public health monitoring [3,4]. Initially developed for military and industrial use, drones have evolved into multifunctional platforms for civil applications, including environmental management, agricultural monitoring, and epidemic response [5]. In vector-borne disease control, drone technology provides a powerful alternative to traditional ground-based surveillance by capturing high-resolution images of large and often inaccessible areas [6]. This remote sensing capability identifies and maps mosquito breeding sites, particularly in densely built urban settings or ecologically sensitive environments where conventional inspections may be difficult, dangerous, or intrusive [4].

The deployment of drones in public health initiatives is also increasingly linked with machine learning and artificial intelligence tools, which enable automated detection of environmental features such as stagnant water containers, vegetation density, or building types associated with vector habitats [7]. These advances have made drones appealing for reactive surveillance during outbreaks, proactive planning, and targeted intervention [8]. However, while the technological promise of drones is widely acknowledged, their successful integration into health systems is far from straightforward [9,10]. As drone use extends into residential neighborhoods and community spaces, their visibility and novelty raise questions about privacy, safety, and trust, particularly in socially and culturally sensitive contexts.

The potential of drone technology is especially relevant in mosquito-borne dengue fever, which remains a major public health concern in tropical and subtropical regions. It's vector, the *Aedes aegypti* mosquito, thrives in urban environments and breeds in artificial containers holding stagnant water. High-resolution aerial imagery collected by drones can support the identification of potential larval habitats, prioritizing hotspots for intervention, and enabling near-real-time decision-making [6]. Drone mapping has demonstrated its usefulness across the globe, including Mexico, where they were successfully used to identify *Aedes aegypti* breeding sites around households [6]. In Brazil, the technology-enhanced detection sensitivity and delivered timely spatial data to support intervention planning (2021); and applied machine learning techniques to drone-acquired imagery to automatically identify potential mosquito breeding containers, significantly enhancing efficiency compared to manual inspections [7]. Integration of drone data into Geographic Information Systems (GIS) and predictive models enhances long-term vector control planning and outbreak forecasting [11]. Yet despite this potential, the actual deployment of drones in public health remains largely limited to pilot studies and proof-of-concept experiments. These efforts are often implemented without the full integration of local knowledge, institutional support, or community input [12,13].

The state of Selangor in Malaysia has consistently battled recurring dengue outbreaks, accounting for 53.2% of national cases between 2015 and 2021. It and presents the highest incidence in the country, with up to 1,115 dengue cases per 100,000 people in 2019 [14,15]. These are driven by rapid urbanization, high population density, inconsistent waste and water management, and climate variability [16]. Traditional control measures such as fogging, larviciding, and community inspections are labor-intensive, resource-consuming, and often ineffective in comprehensively locating the dispersed and dynamic breeding sites of mosquitoes [17]. However, the implementation of new technologies like drone-based surveillance is not only a technical issue, as it is profoundly shaped by the social and cultural context in which it takes place [6]. In traditional or close-knit communities, drones can be perceived as intrusive, unfamiliar, or even threatening, especially when their purpose is poorly understood, or when there is a lack of transparency about their use. Concerns about data privacy, surveillance over private property, and the intentions behind drone usage can overshadow the public health imperative, leading to resistance or disengagement [18]. Without a clear strategy for community engagement, even the most sophisticated technological tools may fail to yield sustained impact or local ownership.

To better understand the practical and social dynamics of implementing drone technology in mosquito habitat surveillance as part of a dengue control strategy, this study explores the perceptions, concerns, and expectations of key stakeholders and community members. It draws on insights from a newly established cohort of households in Selangor. The goal is to inform the implementation of a five-year prospective cohort study and an automated

drone surveillance program in the region. Drawing on interviews with health professionals and residents across eight localities, the study investigates how drone use is framed and understood at the community level, what barriers and enablers shape its acceptance, and how these insights can inform the sustainable integration of drones to implement a five-year cohort study. Currently, drone-based surveillance is not part of Selangor's official dengue control programs. This prospective study was therefore designed to assess community and stakeholder perceptions, acceptance, and social feasibility before launching a larger cohort study. The findings will inform methodological development and support potential integration of drone technology into future pilot vector control programs, particularly to enhance container surveillance efficiency.

## Methodology

### Ethics statement

The Medical Research & Ethics Committee of the Ministry of Health Malaysia approved this study (NMRR ID-24-00374-8SP, 25 March 2024). Formal consent was obtained from all participants, ensuring they understood the study's purpose, procedures, and rights. Participants' responses were anonymized to protect privacy, and informed consent was obtained before data collection.

### Conceptual framework

Our framework (Fig 1) integrates insights from community-based vector control, technological adoption in public health, and drone-assisted surveillance. To structure this, we applied a five-stage logic model: *Inputs, Processes, Outputs, Outcomes, and Impacts*, commonly used in public health intervention planning and evaluation [6,19,20]. This approach is consistent with established implementation frameworks such as RE-AIM and the Consolidated Framework for Implementation Research (CFIR), which emphasize linking resources and activities to measurable outcomes and long-term impacts [21–23].

The framework captures the technical, organizational, and social dimensions essential for sustainability of drone use. It integrates technological elements like drone mapping and AI analytics with community engagement strategies, and institutional collaboration mechanisms. We consider trust, compliance, and adoption in public health interventions [24]. Through the use of our framework we developed tools to probe inputs (resources, infrastructure, community readiness), processes (household participation, institutional roles), and anticipated outputs/outcomes (acceptability, feasibility, and perceived benefits).

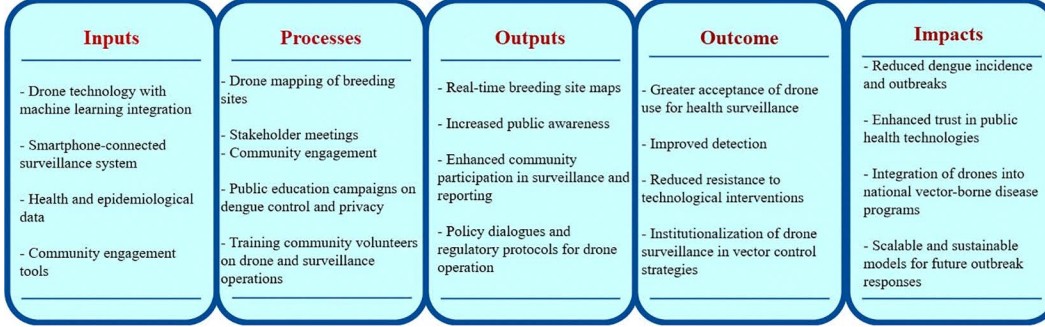

**Fig 1. The research conceptual framework.**

## Study setting

Selangor is located on the west coast of Peninsular Malaysia [25] and it is the most populous and economically developed state (Fig 2). It has a population of over 7 million [25] and its economy is driven by industries such as manufacturing, the service industry, and commerce, making it a vital economic hub for the country. Despite this wealth, Selangor consistently reports more than 60% of Malaysia's dengue cases annually. This is driven by its rapid urbanization, high population density, and high-rise buildings that create ideal breeding grounds for *Aedes* mosquitoes [26]. Compared to other regions in Malaysia, Selangor's communities live in more densely packed urban environments, with high mobility and close living conditions that increase the risk of dengue transmission. These factors complicate efforts to control the disease effectively.

## Data collection

A total of 506 households were initially approached across the eight study localities (targeting approximately 60 per site). Households were identified using a case purposive sampling approach to capture diversity in demographic and socioeconomic characteristics within each locality [27]. The intended respondent was the household head; if the household head was unavailable or declined, an adjacent household was approached as a replacement. In cases where the household head could not participate, another adult member was interviewed. Of the 506 households approached, 480 consented and completed in-depth interviews, corresponding to an overall acceptance rate of 95%.

In this study, the number of in-depth interviews (IDIs) was pragmatically determined by the baseline design of the larger five-year cohort program. Specifically, one respondent per household was interviewed during enrollment, resulting in 480 IDIs across the eight localities (approximately 60 households per site). Thus, the IDI sample size reflects the cohort's baseline recruitment strategy rather than being determined by a priori notion of saturation. Household IDIs lasted 25–40 minutes.

For the household-level in-depth interviews (IDIs), sixty participants were selected from each of the eight localities, ensuring diversity in urban heterogeneity and socioeconomic and educational backgrounds (Table 1). Face-to-face semi-structured interviews were conducted, and audio recorded on smartphones. Participants in the Key Informant Interview (KIIs) were selected using expert sampling [27], targeting individuals with deep knowledge in public health, entomology, health education, and community mobilization. A total of six key informants participated, including two Public Health Medicine Specialists (one from the Ministry of Health's Vector-Borne Diseases Division and one from the Selangor State Health Department), two Health Education Officers, one entomologist, and one COMBI (Communication for Behavioral Impact) leader. The KIIs were designed to provide complementary contextual insights. All stakeholder interviews were conducted via Zoom in May 2024. Each interview ranged from 45 to 60 minutes in length. Topics included perceptions of dengue as a health threat, views on drone surveillance, privacy concerns, expectations of efficiency, and trust in project implementation.

All interviews were conducted by a team of trained public health researchers and eight postgraduate students with prior experience in qualitative research and community engagement. Interviewers underwent structured training that covered the study protocol, ethical considerations, interviewing techniques, probing strategies, and approaches for addressing sensitive issues such as privacy and data use. Standardized training modules and mock interviews were conducted to ensure consistency across interviewers and study sites. Interview guides were developed in English and translated into Bahasa Melayu. All interviews were conducted in Bahasa Melayu to facilitate participant comfort and natural expression. During transcription and translation, bilingual researchers performed quality checks on a subset of transcripts to ensure fidelity and preserve nuanced meanings.

## Data analysis

Thematic analysis was employed for household and key informant interview data, which is useful when looking for recurring patterns [27]. We followed Braun and Clarke's steps for conducting the analysis [28]. The first step was the careful and immersive reading of all transcribed and then translate the interviews to gain a deep understanding of the content.

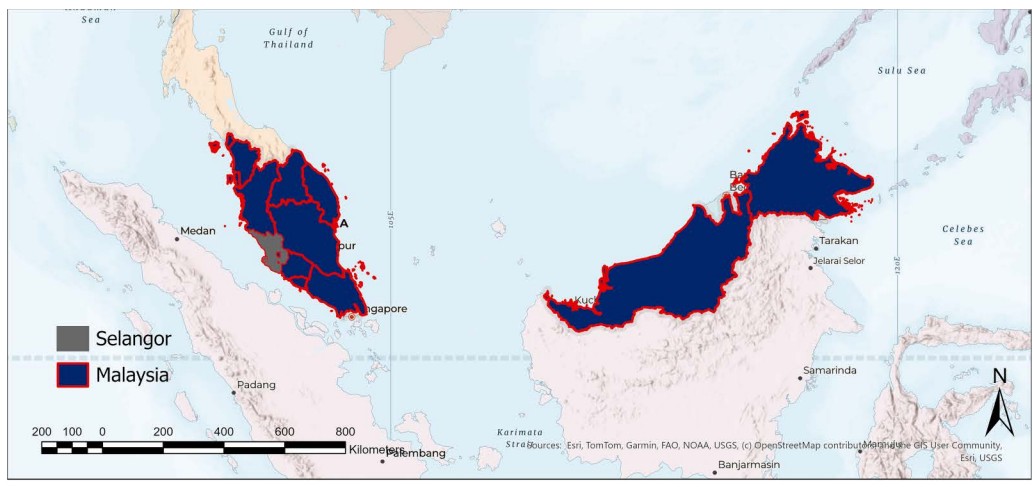

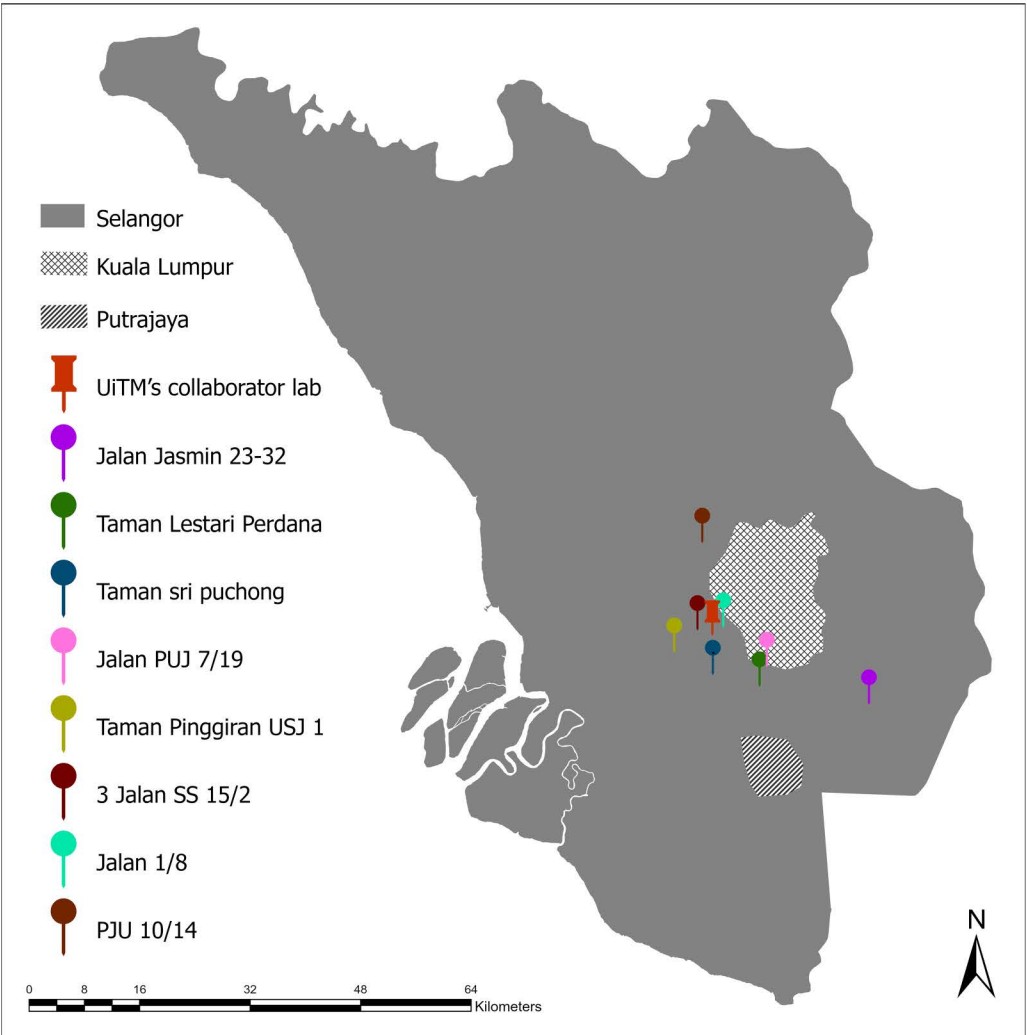

**Fig 2. Study area showing the location of interview sites and dengue intervention zones in Selangor, Malaysia.** Base map contains information from OpenStreetMap and OpenStreetMap Foundation, available under the Open Database License (ODbL). Source: https://www.openstreetmap.org.

Table 1. Distribution of respondents by gender, age group, occupation, and education.

| Respondents | Gender, N (%) | | Age group, N (%) | | | Occupation, N (%) | | | | Education, N (%) | | | |
|---|---|---|---|---|---|---|---|---|---|---|---|---|---|
| | Male | Female | 18-30 | 31-55 | 56 and above | Private sector | Government sector | Self Employed | Not working | Primary school | Secondary school | Diploma and degree | Master's and PhD |
| Stakeholders | 4 (1.85) | 2 (0.74) | | 6 (2.35) | | 2 (1.74) | 4 (1.90) | | | | | 3 (0.99) | 3 (4.55) |
| Community | 212 (98.15) | 268 (99.26) | 146 (100) | 249 (97.65) | 85 (100) | 113 (98.26) | 207 (98.10) | 74 (100) | 86 (100) | 0 (100) | 118 (100) | 299 (99.01) | 63 (95.45) |

This was carried out by ST & UH. Afterwards, an open and iterative coding strategy was conducted manually by ST, ALR & UH. Codes were clustered into categories in a grounded and iterative discussion process between ST, ALR & UH. Codes were then grouped into categories and then these into themes by ST & ALR. Finally, the themes were narrated and illustrative quotes were chosen by ST & ALR.

**Findings**

Table 1 shows that a total of 486 respondents participated in the study, comprising 6 key informants and 480 community respondents. The highest proportion were female (268; 55.1%). The largest age group was 31–55 years, with 249 respondents (51.2%). In terms of occupation, the majority were employed in the government sector (207; 42.6%). For education level, the secondary school category recorded the highest number of respondents (299; 61.5%).

## Theme 1: Community perceives dengue as a persistent health threat

In many communities, dengue is seen not as a temporary outbreak but as a recurring problem that is part of daily life. The majority of our participants associated dengue with familiar environmental factors such as stagnant water and nearby construction sites. There is a sense that, despite ongoing prevention campaigns, dengue continues to resurface. This particularly happens during rainy seasons, reinforcing the idea that it is a permanent and lingering health risk.

The repeated exposure to dengue has made residents more attuned to its symptoms and dangers. Even those who have not personally suffered from dengue often know someone in their community who has. As a result, dengue is treated seriously and is seen not only as a health issue but also as a disruption to daily routines, education, and livelihoods. The threat of illness lingers in the background, influencing behaviors around cleanliness, water storage, and how outdoor spaces are used.

"Dengue is a real problem here. It's not just once in a while. We hear of it almost every few months. You clean your place, but still, someone nearby might have stagnant water. We try, but it keeps coming back. It's worrying, especially for families with small children and the elderly. That's why it makes sense to try when there's something new, like a drone, to help detect the breeding places. But you have to make sure people really understand how it helps." Male community member 04.

This heightened awareness has created a foundation of openness to new surveillance strategies. The communities in our study do not see dengue as something that can be addressed through one-off interventions. Instead, there is a desire for continuous, proactive solutions to supplement their efforts. Our participants framed the drones as a long-term support tool for mosquito control that could be welcomed and seen as complementary to the community's existing vigilance.

## Theme 2: Community expects drone usage to be focused and efficient

There is a clear expectation that using drone container surveillance for dengue control should be strategic and purposeful. Community members are particularly interested in how drones can detect mosquito breeding sites that are otherwise difficult to reach or overlooked in manual inspections. Rather than random surveillance, people expect drones to be deployed with specific objectives, such as monitoring known hotspots.

Residents view efficiency as a sign of professionalism and seriousness. It was expressed that if drones were seen operating in repetitive or unfocused ways, it would generate skepticism about the project's effectiveness. There was also a concern that excessive drone activity might become intrusive or cause fatigue over time. Suggestions like including a well-structured and well-communicated implementation plan can help address these concerns and demonstrate that the technology is being used in a way that is both responsible and impactful.

"I think it's a good idea if the drone is used in a very targeted way. Don't just fly around randomly and go to places where there are usually water collection problems or areas with dengue cases. That makes people feel like the drone is being used smartly, not just to show off. And if we know it will come every Monday at 2:30 pm, as you said, that schedule helps us feel prepared, and we can even help the team by pointing out problem spots."

The community's support for drone use was often framed around its ability to enable rapid detection of breeding habitats to guide a targeted public health response and so, the community is more likely to perceive it as a valuable innovation. Participants reported being more inclined to support strategies for done use that show clear benefits, especially when these align with their own understanding of how dengue spreads. By emphasizing how drones contribute to faster, more accurate responses from health authorities, the project can meet community expectations and gain deeper trust.

"The biggest issue is privacy. We must ensure the community knows when and where the drone is flying. The operator must also be trained, and the footage must be stored securely. Because once people think you're spying, they won't care if it's for dengue control. They will just resist. So, you need to put rules, like SOPs (clear guidelines), and communicate them. Maybe also work with NGOs to educate people on what the drone is and isn't doing. That way, trust can be built even before the drone is in the air." Female community member 150)

At the same time, a few participants expressed doubts about whether drones could be truly effective in all environments. There were concerns about the limitations of drone access in densely built-up or high-rise areas and skepticism about how comprehensive the coverage could be. Some questioned whether drones could detect small breeding sites or stagnant water hidden in cluttered spaces. These views highlight the need to manage expectations and communicate the technology's strengths and limits.

"Honestly, I'm not sure how well the drones can work in all these areas. Maybe in open spaces or landed homes, yes, but what about apartments or flats? Some people who live in the apartments can use the drone to see small containers or buckets on their balconies. Even in ground-level houses, people have backyards full of clutter, broken furniture, and hidden spots." Male community member 112

## Theme 3: Community views drone surveillance as non-invasive

While surveillance often raises concerns in other contexts, most of our participants from the community view drone-based monitoring as non-threatening, particularly when it is clearly explained. Because the focus is on environmental conditions rather than personal spaces, participants reflected that they are less likely to associate drones with privacy violations or

unwanted surveillance. This acceptance depends on how well the project distinguishes itself from other types of drone use that might be perceived as intrusive.

When community members understand that drones only collect data on water accumulation, container locations, or potential mosquito habitats, more will reportedly become more comfortable with the idea. This would happen quicker if a community is informed about the limits of the technology and when there is transparency about how the collected data will be used. Clarity and education about the drone's role are critical to maintaining this non-invasive perception.

> "I'm comfortable with it because it's not looking at people. It is not for spying or recording private things. It is just checking the environment, like where there might be standing water or containers that could bring mosquitoes. If you explain clearly that the drone is only detecting breeding sites, not watching people, most of us are okay with it. But you must be open and tell us what it can see, what it cannot, and who will use the information. Male community member 04

This acceptance, however, is conditional. If, at any point, residents begin to feel that drones are being used excessively or without proper explanation, the perception could shift. Maintaining a sense of respect and boundaries, alongside regular updates, will be essential. Framing drones as tools for environmental protection rather than surveillance of people reinforces a sense of safety and aligns with the community's existing efforts to eliminate mosquito breeding sites.

Nonetheless, a few individuals expressed unease, particularly regarding drones flying close to homes or in residential areas with minimal clearance. Some feared that even with reassurance, flying devices near windows could create discomfort or raise suspicions among elderly residents or those unfamiliar with the project. These concerns signal that sensitivity around privacy, while not widespread, does exist and should be addressed through careful planning, clear messaging, and possibly route adjustments when needed.

> "Some people, especially the older ones, might not feel comfortable if the drone flies too close to their windows. Even if you tell them it's for dengue control, they might still wonder, 'Why is it coming here? Is it filming us?' You know, not everyone understands technology or trusts it right away. So, it's not about rejecting the project; it's just... you need to be careful where you fly and maybe avoid flying too low near bedroom windows or places where people feel exposed. It is better if there's a way to let people know in advance or even adjust the route slightly to avoid sensitive spots." Key informant 1

## 4. Motivations to participate are rooted in communal benefit

People's willingness to support and participate in the project is often driven by collective responsibility rather than personal gain. Many participants described dengue prevention as a shared community responsibility. They are motivated by the desire to protect their own households and their neighbors, especially the elderly and young children. This orientation toward community well-being shapes how individuals respond to new strategies.

There is also a sense of pride in being part of something that could benefit the broader public. Contributing to a project to reduce disease across the neighborhood gives people a sense of purpose and involvement in something larger than themselves. This motivation is particularly strong in communities that have previously participated in public health campaigns, where there is already a culture of engagement and cooperation.

> "From my own perception... most of the people who are staying here already are aware that this is an issue. So, I believe that with some explanation, telling them the benefits of the study since it will benefit the environment and the community, once you approach them and explain to them in detail what the research entails, they will be willing to participate. Because it will help eradicate many things in the community. You just have to go to places like the playground or cafeteria, not just approach people randomly on the road. That way, people will feel more respected and be more open." Female community member 150

"I think for this kind of case, if you want to be successful, you need to create an NGO, or you can collaborate with the government, and if we are advertising, all people understand this kind of project is in the process. And because this is one of Malaysia's biggest public health problems, I think people like to implement this dengue project." Key informant 4

Encouraging and sustaining this collective spirit is perceived as key to long-term success. In the interviews, many mentioned feeling rewarded when they see their participation in studies like the one presented here makes a difference for other communities. They also expressed feeling recognized as partners in the effort, they are more likely to remain engaged. Simple gestures like providing updates, seeking community input, or acknowledging contributions can reinforce this communal mindset and foster deeper collaboration throughout the project.

Although some residents mentioned incentives, their expectations were often modest and not focused on direct material rewards. Most participants emphasized non-direct incentives as more meaningful, such as gaining knowledge about dengue prevention, being included in decision-making processes, or seeing visible improvements in their community. These forms of recognition and engagement were valued more than gifts or payments. While some participants mentioned that small tokens or compensation might encourage initial participation, the prevailing attitude was that involvement should come from a shared desire to protect the community rather than personal benefit. This highlights the importance of framing participation as a collective action and valuing intrinsic motivations over transactional ones.

## Discussion

This study revealed a strong willingness among Selangor community members in dengue-endemic areas to participate in drone-based surveillance, and all participants expressed some degree of openness to such strategies. These findings contribute new insights into how drone technology intersects with community health needs, public trust, and social responsibility and highlight key dimensions that must be navigated for such innovations to succeed: the everyday perception of disease risk, expectations of operational effectiveness, the navigation of privacy concerns, and the collective motivations behind community engagement. As previous research on vector control has shown, trust-building, community education, and participatory engagement are critical for the acceptance and success of health interventions regardless of their technical merit [29–31].

The community's characterization of dengue as a chronic, recurring burden rather than an episodic threat offers an important context for understanding their receptiveness to new technologies. The normalization of dengue risk, especially during the rainy season, has led to a deeply personal and collective memory of the disease. This perception has generated an openness to innovation, particularly with solutions that are seen as proactive, continuous, and environmentally embedded. Studies in Latin America and Southeast Asia have shown similar dynamics, where endemic exposure to mosquito-borne illnesses increases the community's desire for sustained interventions beyond reactive spraying or health education campaigns [4,6].

In settings where dengue is highly prevalent, drone technology is perceived not merely as novel but as potentially complementary to residents' efforts to mitigate mosquito breeding. However, this sense of promise is tightly linked to how technology is implemented. Participants repeatedly emphasized that the success of drone surveillance hinges on strategic deployment. They expect the drones to serve precise purposes, such as targeting difficult-to-access areas or hotspots identified by case clustering rather than random or symbolic flights. Similar concerns have been echoed in the literature on smart health technologies, where the perceived alignment between technological capability and community-identified need is key to sustaining legitimacy [11].

Efficiency is both a technical and communicative matter. When community members observe well-coordinated, data-driven deployment of drones, it reinforces trust in the broader intervention. On the contrary, repetitive or unfocused use can fuel skepticism [32]. Concerns about operational constraints, such as drone visibility in high-rise housing or detection limitations in cluttered or shaded environments, signal a sophisticated community understanding of the strengths and

limitations of drone-based surveillance. Managing these expectations requires transparent public education that acknowl-edges what drones can realistically deliver, aligning public perception with operational scope [33]. Having Standard Operating Procedures (SOPs) maintains trust and establishes a clear communication infrastructure by engaging NGOs and local leaders to disseminate accurate information. NGOs play a critical bridging role by translating technical objectives into culturally accessible terms, managing expectations, and reinforcing the ethical boundaries of surveillance [18,30].

Despite scholarship associating surveillance technologies with loss of privacy or intrusion [34], this study reveals that when drones are clearly positioned as tools for environmental monitoring, communities can distinguish them from forms of surveillance that feel threatening. Participants clarified that their comfort depended on consistent communication and guarantees about the nature of collected data. They preferred that drones fly in open areas and avoid proximity to private residential windows, particularly in communities with elderly residents or those unfamiliar with technology. These findings highlight that health technologies are accepted more readily when introduced within a framework of ethical transparency, routine communication, and reciprocal feedback [29,31].

Material incentives did not primarily drive participation in the drone initiative. Instead, respondents expressed strong communal motivations. When projects are grounded in the local community's values, many community members will see their involvement as part of a shared responsibility for safeguarding vulnerable children, the elderly, and those with pre-existing health conditions [35]. In other words, matching local values can harness the power of the community for more than just acceptability. In this context, participation was tied to social value and mutual care rather than individual benefit [36]. Similar patterns have emerged in community-based dengue interventions elsewhere, where co-benefits such as cleaner environments, neighborhood recognition, and educational empowerment drive long-term engagement [30].

Moreover, being part of a public health innovation generated a sense of civic pride. People wanted to be seen as proactive contributors to national or global challenges. This is in line with a study in Guatemala, where civic values played a key role in implementing health-related outreach activity. While some modest incentives were appreciated, participants valued inclusion in decision-making, feedback on intervention results, and impact visibility [37]. This is consistent with global literature on citizen participation in health surveillance, where recognition and belonging are often more powerful than compensation in sustaining grassroots involvement [31].

Our findings support the argument that public health technologies such as drones cannot be detached from the social systems in which they operate. The success of such technologies is co-produced through technical capability, policy integration, and social negotiation. This includes community participation programs rooted in shared power and decision-making, where communities are active partners [38]. Governance, or how drones are rolled out due to social and political processes, involves many relevant stakeholders that navigate complex settings and follow their own agendas. Project designers must prioritize clarity, co-design strategies, feedback mechanisms, and ethical safeguards for sustained implementation. Technologies should not only be deployed but also explained, situated, and made accountable to those who live under their presence.

While this study found high community acceptance in Malaysia, we recognize that successful, long-term implementation of this policy would require a comprehensive operational framework. Therefore, for a future study or pilot program to proceed, it will be critical to develop and transparently communicate a Standard Operating Procedure (SOP). This SOP must address key governance and privacy issues, including securing necessary approvals from aviation authorities, establishing clear data governance protocols for image storage and access, and operationalizing a consistent community notification system. Addressing these practical concerns will be essential for translating the high acceptance found in this study into sustained public trust and participation.

## Strengths and limitations

This study's design captures perceptions at one time point, limiting insights into evolving attitudes during implementation. Findings are context-specific to eight urban localities in Selangor and may not generalize elsewhere. Variability in

technological literacy may shape responses, and perceptions are hypothetical without observing actual drone use. Manual thematic analysis enabled in-depth, context-rich insights, though consistency in theme identification relied on close collaboration within a small team.

This study was conducted within the socio-cultural context of multi-ethnic, urban communities in Selangor, Malaysia, where perceptions and high acceptance were shaped by local norms and strong trust in public health authorities. While these dynamics are context-specific, the findings provide valuable insights that can guide adaptation of drone-based surveillance approaches in diverse settings with different community–government relationships and views on technology.

While drones identify potential rather than confirmed breeding sites, subsequent field validation ensures accurate targeting of control measures. This two-step process improves efficiency by directing vector control teams to likely hotspots. Moreover, the geospatial data on container types and density provide valuable inputs for predictive modeling, enhancing the accuracy of spatio-temporal forecasts and supporting more proactive dengue prevention strategies.

## Conclusion

This study highlights how communities in Selangor perceive dengue as an enduring health challenge that disrupts daily life and livelihoods, reinforcing the need for continuous and proactive interventions. Residents demonstrated heightened awareness of the disease and recognized the importance of sustained surveillance, which created openness toward innovative approaches such as drone-based monitoring. Importantly, participants emphasized that drones should be deployed strategically and efficiently, targeting high-risk areas rather than conducting unfocused flights, and that clear communication of their purpose and limitations is essential to maintain trust. While most viewed drone surveillance as non-invasive when focused on environmental factors, some concerns about privacy and discomfort near residential spaces indicate the importance of transparent guidelines, operator training, and community engagement.

Support for participation was largely motivated by collective responsibility and a desire to protect vulnerable groups, rather than personal incentives. Together, these insights underscore the value of aligning new technologies with community expectations, ensuring respectful engagement, and reinforcing shared responsibility in dengue prevention efforts.

## Acknowledgments

We thank all eight postgraduate students at Universiti Putra Malaysia for assisting in the data collection.

## Author contributions

**Conceptualization:** Rahmat Dapari, Timothy C. Guetterman, UBYDUL HAQUE.

**Data curation:** Rahmat Dapari.

**Formal analysis:** Rahmat Dapari, Safiyeh Tayebi, Ana Lorena Ruano.

**Funding acquisition:** UBYDUL HAQUE.

**Investigation:** Rahmat Dapari, Ana Lorena Ruano, UBYDUL HAQUE.

**Methodology:** Rahmat Dapari, Safiyeh Tayebi, Ana Lorena Ruano, Timothy C. Guetterman.

**Project administration:** UBYDUL HAQUE.

**Resources:** UBYDUL HAQUE.

**Software:** Safiyeh Tayebi.

**Supervision:** Ana Lorena Ruano.

**Validation:** Rahmat Dapari, Safiyeh Tayebi, Ana Lorena Ruano.

**Visualization:** Safiyeh Tayebi.

**Writing – original draft:** Rahmat Dapari, Safiyeh Tayebi.

**Writing – review & editing:** Rahmat Dapari, Safiyeh Tayebi, Ana Lorena Ruano, Timothy C. Guetterman, Seok Mui Wang, Siti Hafizah AB Hamid, Sohel Rahman, Jürgen Pilz, Nazri Che Dom, UBYDUL HAQUE.

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
