## [Decision Letter · Decision Letter 0]

2 Sep 2025

Community Participation and Technological Innovation: A Prospective Cohort Study on Drone-Based Dengue Surveillance in Malaysia

Dear Dr. HAQUE,

Thank you for submitting your manuscript to PLOS Neglected Tropical Diseases. After careful consideration, we feel that it has merit but does not fully meet PLOS Neglected Tropical Diseases's publication criteria as it currently stands. Therefore, we invite you to submit a revised version of the manuscript that addresses the points raised during the review process.

Please submit your revised manuscript within 60 days Nov 01 2025 11:59PM. If you will need more time than this to complete your revisions, please reply to this message or contact the journal office at plosntds@plos.org. Please include the following items when submitting your revised manuscript:

We look forward to receiving your revised manuscript.

Kind regards,

Lawrence Mugisha, PhD

Academic Editor

Nigel Beebe

Section Editor

Shaden Kamhawi

co-Editor-in-Chief

Paul Brindley

co-Editor-in-Chief

**Journal Requirements:**

At this stage, the following Authors/Authors require contributions: Rahmat Dapari, Safiyeh Tayebi, Ana Lorena Ruano, Timothy C. Guetterman, Seok Mui Wang, Siti Hafizah, Sohel Rahman, Jürgen Pilz, Nazri Che Dom, and UBYDUL HAQUE. Please ensure that the full contributions of each author are acknowledged in the "Add/Edit/Remove Authors" section of our submission form.

4) Tables should not be uploaded as individual files. Please remove these files and include the Tables in your manuscript file as editable, cell-based objects. For more information about how to format tables, see our guidelines:

https://journals.plos.org/plosntds/s/tables

Potential Copyright Issues:

- Figure 2. Please (a) provide a direct link to the base layer of the map (i.e., the country or region border shape) and ensure this is also included in the figure legend; and (b) provide a link to the terms of use / license information for the base layer image or shapefile. We cannot publish proprietary or copyrighted maps (e.g. Google Maps, Mapquest) and the terms of use for your map base layer must be compatible with our CC BY 4.0 license.

6) In the online submission form, you indicated that "All data can be obtained from the corresponding author". All PLOS journals now require all data underlying the findings described in their manuscript to be freely available to other researchers, either

- In a public repository

- Within the manuscript itself

- Uploaded as supplementary information.

**Reviewers' Comments:**

Reviewer's Responses to Questions

**Key Review Criteria Required for Acceptance?**

**Methods**

-Are the objectives of the study clearly articulated with a clear testable hypothesis stated?

-Is the study design appropriate to address the stated objectives?

-Is the population clearly described and appropriate for the hypothesis being tested?

-Is the sample size sufficient to ensure adequate power to address the hypothesis being tested?

-Were correct statistical analysis used to support conclusions?

-Are there concerns about ethical or regulatory requirements being met?

Reviewer #1: Qualitative methods are a simple way of exploring community feedback and hence by its nature not quatitative

Reviewer #2: (No Response)

Reviewer #3: This article offers interesting evidence from a qualitative sense on the households and stakeholder perceptions of drone-based dengue control in Selangor. Whilst, the findings from the study indicated and discussed on substantial issues and concerns, however, there are some aspects of this article in need of further clarification to ensure its logical flow and consistency. Here are some suggestions for the authors kind consideration:

- Abstract

there is a need to include the key word ‘prospective cohort study’ in the Background statement of: “To inform a five-year automated…’ as mentioned in the article for consistency.

Also, in the conclusions section, there is a thematic like analysis done that highlighted community motivation, collective responsibility, …trust, transparency, and consistent communication…and the success of the project. Please ensure consistency of what is written in the abstract to be similar to what is described in the article sections.

- Introduction

There are many statements made in the article that are not supported with the relevant references, for instance in pg. 4-5:

‘….technological promise of drones is widely acknowledged, their successful integration into health systems is far from straightforward…..”

‘…often implemented without the full integration of local knowledge, institutitional support, or community input.’

‘Selangor, Malaysia, has consistently battled recurring dengue outbreaks’

Statements that are based on factual instances need to have statistical evidence together with the relevant references.

- Methodology

In the conceptual framework, it was stated that the framework integrates ‘…insights from the literature on community-based vector control, technological adoption in public health, and drone-assisted surveillance. Where is the summary of literature review conducted for this article?

As for Figure 1, the research conceptual framework, was it modified or adapted from other sources? Please indicate if so.

Similarly for the study setting, statement that are factually driven such as ‘…rapid urbanisation, high population density and high-rise buildings that create ideal breeding grounds for Aedes mosquitoes..’ should be supported with references.

In Figure 2, the study setting, map given is difficult to read and see the exact location. Consider to revise this map.

There was no justification as to why the eight localities were chosen to represent Selangor. And there is usage of acronyms that are not explained before used, such as IDI, KII, ST, UH, ALR. Consider to provide the explanation.

In terms of the layout, the writing and empty spaces in between each section, as observed from Data Analysis in pg. 8 and page 9 Findings. Please rectify accordingly.

**Results**

-Does the analysis presented match the analysis plan?

-Are the results clearly and completely presented?

-Are the figures (Tables, Images) of sufficient quality for clarity?

Reviewer #1: Please provide more context for the readers on what communities you are referring to. Malaysian communities? Findings will be relevant to the culture in which they are being studied, so this needs to be emphasised throughout, especially in the discussion. It is also a limitation to findings which needs to be discussed in the limitations section.

Page 10: please explain in further detail how drones can be deployed with the specific objective of “identifying new clusters based on reported dengue cases” or otherwise remove.

Page 10: You mention drone use “framed around rapid detection and targeted response”. The majority of what has been discussed to this point has been around surveillance. In this context, what do you mean by rapid detection and response?

Reviewer #2: (No Response)

Reviewer #3: As for this results section, the qualitative style is not fully explored, often the authors seem convenient to use expression such as ‘many in the community appear to view….’. There is a need to be objective and specific even though you are describing the findings in a qualitative manner. Since the authors collected a household’s data of 506 respondents, how was the demographic results as stated in Table 1 assisted in the presentation of this Findings section? In fact, the authors seem to give a blanket statement of findings for the eight localities. Even though, the author did mention in pg. 7 ‘ensuring diversity in urban heterogeneity and socioeconomic and educational backgrounds.’ However, there were no clear attempt made by the authors to provide this diversity in terms of the households and stakeholders perceptions and acceptance towards informing them on the implementation of this five-year automated drone surveillance program.

Whilst, it is interesting when the authors used quotes from the respondents and key informants to support each main sub-section of the findings. It is still important to address the importance of eight localities in highlighting what were the findings in terms of its similarities and differences. The authors should also incorporate 1-2 paragraph before presenting each of the sub-sections as stated there are four sub-sections, what are they? Themes? Elements? Dimensions? Unclear.

Also, there was no specific mention to the program of this prospective cohort study in detail, particularly to describe the five-year automated plan? What is and who is involved? These pertinent points were not mentioned. How was the prospective cohort study conveyed to the households? Missing points to provide logical understanding as to how the findings were collected from the households and stakeholders.

**Conclusions**

-Are the conclusions supported by the data presented?

-Are the limitations of analysis clearly described?

-Do the authors discuss how these data can be helpful to advance our understanding of the topic under study?

-Is public health relevance addressed?

Reviewer #1: Use of Terminology

Throughout the manuscript the authors continually refer to the use of drones for dengue control or dengue surveillance. It is even in the title of the manuscript. The drones being described in this manuscript and references provided are not being applied in dengue control, they are being employed in container surveillance for mosquito vectors of dengue (sometimes referred to as mosquito breeding site monitoring in the manuscript). It is important that this difference is made clear throughout the majority of the manuscript and removed from the title. The only way drones can be talked about in the context of dengue, is to improve outcomes of dengue management or dengue control (ie. improved efficiency of surveillance improves management outcomes).

Use of the term drone interventions and drones as mosquito control tools implies that the drones are acting to manage dengue, when in the case described in the manuscript or in the references they are being used for passive monitoring. Again, please consider the use of a passive term such as surveillance.

The use of the wording “potential breeding sites” is critical in this manuscript, as drones are not as useful as active monitoring when samples are taken from containers for identification. Potential mosquito breeding sites still need to be validated by field staff on the ground. Detection of potential breeding sites will be more efficient in dengue endemic areas to direct mosquito control but not for those outside endemic regions that do not utilise insecticide regular spraying programs.

Please include the limitation that findings may only be applicable to the cultural setting they were observed in.

Reviewer #2: (No Response)

Reviewer #3: - Discussion

Having a clear explanation to what the sub-section described in the findings would help to reiterate in the discussion although there were mentioned of them as ‘key dimensions. Please clarify on this. Also, there are vague statements made in pg. 14 where authors mentioned ‘As previous research on vector control has shown’…which one?

- Conclusion

In this section, the authors summarized what has been said in the findings and discussion sections, perhaps the authors could consider to synthesize the main points and how to bring these points forward from the study.

**Editorial and Data Presentation Modifications?**

Reviewer #1: Figure 2: fonts are too small to read.

First use of Aedes aegypti in full then use Ae. aegypti both with italics.

Discussion

Second paragraph needs the font corrected and integrated into one of the other paragraphs before or after this one sentence.

Reviewer #2: Formatting: Remove the unexpected page break after the Introduction; ensure consistent section headers. Additionally, there are no line numbers in the pages, as such I'm not providing specific line changes, since that would have taken too much time to reference (page, paragraph and line within paragraphs).

Demographics table: The current Table 1 appears garbled; please present as a clean table with clear headers and totals (n and %).

References: Replace Wikipedia citations (e.g., for Selangor) with official government or statistical sources. Check that all claims in the framework and discussion cite primary literature (several sentences mention studies but list only a single reference). There are only 26 references, almost as few as a short report (20 max).

Figures: Ensure vector exports for diagrams and color-blind-safe palettes (e.g., viridis-family) where possible.

Reviewer #3: (No Response)

**Summary and General Comments**

Reviewer #1: The manuscript “Community Participation and Technological Innovation: A Prospective Cohort Study on Drone-Based Dengue Surveillance” applies a qualitative approach to explore householder and stakeholder perceptions of drone-based surveillance in Selangor. Authors conducted semi-structured interviews with 480 households to elucidate common themes on the use of these drones in residential areas.

The manuscript is well written and the outcomes of the interviews are clear and findings important for improving the future efficiency of container surveillance in areas where dengue is endemic. The authors emphasise that drone-based surveillance is defined by local social and cultural context which is an important finding for these types of potentially invasive technologies. While I can find no major issues with the manuscript, I would like the authors to reconsider some of their language to ensure the reader is not confused about the use of drones for container surveillance.

While there are no lines that I can directly reference in the manuscript to edit, I will use general terms for where I think improvements can be made.

It would benefit the manuscript for readers to know more about the program under which the drones were being used for surveillance in Selangor. What are the objectives of the larger study? To improve the efficiency of mosquito control? Is container surveillance being validated like was done in the studies referenced?

Reviewer #2: Overall assessment

This manuscript presents formative qualitative work on community acceptance of drone-assisted dengue surveillance in Selangor, Malaysia. The topic is timely and relevant to PLOS NTD. However, there are important issues with study framing (title/design mismatch), methodological reporting (sampling, analysis rigor, and timelines), figure quality/citation, and data availability that should be addressed to meet journal standards. A major revision is required.

Major comments

1. Title/design mismatch (prospective cohort vs qualitative cross-sectional)

Issue: The title and some sections describe a “prospective cohort study,” yet the methods and results report semi-structured interviews (IDIs/KIIs) at a single timepoint with thematic analysis. Re-title and reframe as a qualitative formative study nested within or to inform a planned five-year cohort/surveillance program. You need to clearly state that these findings are qualitative baseline perceptions rather than cohort outcomes.

2. Sampling and recruitment require much greater clarity

Issue: You state that 506 households were approached and 480 IDIs completed (60 per locality × 8). You also mention “typical case sampling” and that adjacent households were contacted if a head was unavailable. It is unclear whether participation was random, purposive, convenience, or a hybrid; who exactly was eligible/responded (e.g., “household head” vs “any adult”); and what inclusion/exclusion criteria were used.

I would recommend a more transparent sampling and recruitment subsection: define sampling strategy (e.g., typical case purposive sampling), sampling frame, approach procedures, respondent definition, and replacement rules. State who was interviewed in households (household head vs any adult), the number approached, refusals, and the acceptance rate per site.

More importantly there is no explanation for why 480 IDIs and 6 KIIs were appropriate. For qualitative work, sample size is typically justified by saturation rather than power. Please provide a clear definition of why this samples sizes were enough. If pragmatic factors dictated the number (e.g., one per household in a baseline cohort), say so, and discuss implications. For KIIs, justify why n=6 was sufficient across stakeholder domains.

3. Timelines and interview duration inconsistent

Issue: KIIs are reported as 45–60 minutes and then 25–40 minutes in the next sentence; household IDIs timelines are not clearly specified (May–July 2024 is later provided). Harmonize and report exact fieldwork windows for IDIs and KIIs and consistent duration ranges for each. Provide the date range for household interviews, not only KIIs.

4. Conceptual framework needs citations and positioning

Issue: The second line of the Conceptual Framework section refers to a logic model without citing foundational frameworks; the five-stage “inputs→impacts” model is common but currently under-referenced. Add appropriate citations to established implementation/logic-model frameworks (e.g., logic models used in public health interventions, implementation frameworks). Also clarify how this framework guided instrument design, analysis, and interpretation (not just presented as a figure). The introduction feels as more references should be added, since the author mentions in several occasions "studies" but sometimes do not provide a citation or just cite 1 study.

5. Qualitative methods rigor (trustworthiness) needs strengthening

Issue: Thematic analysis is described, but reporting is thin on reflexivity, coder training, intercoder agreement/consensus procedures, codebook development, data management, and steps taken to enhance credibility (member checking, triangulation), dependability (audit trail), and confirmability. Report according to COREQ or SRQR (PLOS encourages established qualitative reporting standards). Add: interviewer backgrounds/training, language/translation and back-translation procedures, how the codebook was developed/refined, how discrepancies were resolved, whether saturation was assessed (and how) for the very large number of IDIs (n=480), and any triangulation or member-checking conducted.

7. Analytic transparency and use of descriptive data

Issue: Findings are presented narratively with quotes, but readers of PLOS NTD will expect at minimum participant characteristics (counts/percentages) and, where appropriate, careful indication of how prevalent key themes were (without over-quantifying). Provide a clean Table 1 with respondent demographics by site (n, %; e.g., gender, age bands, occupation, education) and KII roles. In the Results, replace vague frequency terms (e.g., “frequently”, “many”) with n/N (%) where defensible (e.g., “privacy concerns were raised by 42/480 [8.8%] respondents”), while remaining faithful to qualitative norms.

8. Data availability statement (PLOS policy)

Issue: “All data can be obtained from the corresponding author” does not comply with PLOS’ data policy for published articles. Deposit a minimal de-identified dataset suitable for qualitative research (e.g., codebook, theme matrix, anonymized quote compendium, and de-identified participant characteristics) in a stable repository (e.g., Dryad, Zenodo, figshare) and update the Data Availability Statement accordingly.

9. Governance and regulatory context

Issue: You discuss trust and SOPs but provide little about local regulatory approvals for drone flights (aviation/privacy) and data governance (who stores, who accesses, retention, anonymization). Add a short paragraph on operational approvals, data security, image handling (faces/license plates blurred or not recorded), altitude limits near residences, and how community notification is operationalized (e.g., posted schedules).

Limitations: Add potential social desirability bias, translation/interpretation limitations, and context-specific generalizability.

Minor comments

Please review that all acronyms are being spelled out prior to use, specifically for KII.

Figure quality, accessibility, and map/satellite citation: Figures are pixelated; Figure 2’s inset map is too small; color-blind accessibility is not addressed; and any satellite/aerial basemaps must be properly cited (Google/Bing/OSM have explicit requirements).

Household ‘survey’ vs interview terminology: The manuscript sometimes calls the household component a “survey” but methods describe semi-structured IDIs. Use consistent terminology (e.g., “household in-depth interviews”). If a structured questionnaire was also administered, describe its content, type (closed/open items), and how those data were analyzed (basic descriptives), or remove “survey” language.

Reviewer #3: In general, the article significant findings to support the importance of aligning technological innovation with local social dynamics. However, there are parts as stated in the report that need further depth of analysis and clarifications for logical consistency.

PLOS authors have the option to publish the peer review history of their article (what does this mean? ). If published, this will include your full peer review and any attached files.

**Do you want your identity to be public for this peer review?** For information about this choice, including consent withdrawal, please see our Privacy Policy .

Reviewer #1: No

Reviewer #2: No

Reviewer #3: No

**Figure resubmission:**
---

## [Editor Report · Decision Letter 1]

28 Jan 2026

Dear Dr Dapari,

We are pleased to inform you that your manuscript 'Community Participation and Technological Innovation: Baseline Qualitative Insights to Inform a Five-Year Cohort on Drone-Based Dengue Surveillance in Malaysia' has been provisionally accepted for publication in PLOS Neglected Tropical Diseases.

Best regards,

Lawrence Mugisha, PhD

Academic Editor

Nigel Beebe

Section Editor

Shaden Kamhawi

co-Editor-in-Chief

Paul Brindley

co-Editor-in-Chief

---

## [Editor Report · Acceptance letter]

Dear Dr HAQUE,

We are delighted to inform you that your manuscript, "Community Participation and Technological Innovation: Baseline Qualitative Insights to Inform a Five-Year Cohort on Drone-Based Dengue Surveillance in Malaysia," has been formally accepted for publication in PLOS Neglected Tropical Diseases.

Best regards,

Shaden Kamhawi

co-Editor-in-Chief

Paul Brindley

co-Editor-in-Chief
